# TENSORIZED ATTENTION MODEL

## ABSTRACT

In recent years, attention mechanisms have played a crucial role in the success of Transformer models, as seen in platforms like OpenAI's ChatGPT. However, since they compute the attentions from relationships of one or two objects, they fail to effectively capture multi-object relationships in real-world scenarios, resulting in low prediction accuracy. In fact, they cannot calculate attention weights across diverse object types, like 'comments,' 'replies,' and specific 'subjects,' which naturally constitute conversations on platforms like Reddit, representing relationships in real-world contexts. To overcome this limitation, we introduce the Tensorized Attention Model (TAM), which leverages Tucker decomposition to calculate attention weights across various object types and seamlessly integrates them into the Transformer encoder. Evaluations using Reddit and TweetQA datasets, which take into account relationships among various object types, demonstrate that TAM significantly outperforms existing Transformer-based methods in terms of accuracy in response selection tasks.

## 1 INTRODUCTION

The Transformer architecture (Vaswani *et al.*, 2017), renowned for its robust attention mechanism, has achieved extraordinary success in a wide range of natural language processing (NLP) tasks, including named entity recognition (Yan *et al.*, 2019), sentiment analysis (Wang *et al.*, 2020), and machine translation (Wang *et al.*, 2019). Especially in the field of NLP, BERT (Devlin *et al.*, 2019) represents a pivotal milestone, showcasing the remarkable progress achieved primarily through the potency of attention-based mechanisms pre-trained on massive datasets. This model's impact has accelerated the development of numerous other attention-centric models, effectively broadening the horizons of NLP research. A prime example in the realm of attention-based architectures is Chat-GPT (OpenAI, 2023), a prominent generative chatbot with support from InstructGPT (Ouyang *et al.*, 2022), which specializes in generating human-like responses and instructions. Other significant contributions come from Google's Lambda architecture (Thoppilan *et al.*, 2022) and Meta's Llama2 (Touvron *et al.*, 2023). Together, these models underscore the transformative impact of attention mechanisms in elevating the state-of-the-art in NLP.

Current attention-based models typically concentrate on relationships involving just one or two objects, such as query and memory objects, commonly seen in self-attention or source-target attention mechanisms (Vaswani *et al.*, 2017). However, real-world interactions often encompass more complex relationships that involve three or more objects. For example, two connected sentences in a Wikipedia article share a common topic, tweets may trigger a chain of replies connected to specific locations, and multiple utterances in Reddit can stem from a shared conversational context. Incorporating these intricate multi-object relationships has the potential to unlock the latent capabilities of transform-based models further.

The effective representation of such multi-object relationships can be achieved using tensors (Nakatsuji *et al.*, 2016). However, existing methods do not utilize tensors to represent multi-object relationships. For example, (Ma *et al.*, 2019) introduced a Tensorized Transformer model that incorporates a specialized self-attention encoder layer known as Multi-linear attention, along with the Block-Term Tensor Decomposition (BTD) technique (De Lathauwer, 2008). This approach alleviates the computational burden by compressing the extensive parameter set in multi-head attention into a set of 3-order tensors through low-rank approximation. However, it faces several limitations. Firstly, it cannot handle relationships among more than two types of objects due to constraints imposed by their tensor decomposition; it requires equal lengths of query and memory. This constraint

limits its applicability in scenarios involving more diverse object types. Secondly, the approach directly computes transformer output from attention weight tensors and thus is inadequate to obtain transformations from source object to different type target object, making it less versatile. Additionally, the method incurs higher memory overhead compared to traditional Tucker decomposition (Tucker, 1966c; Li *et al.*, 2017) because it simultaneously decomposes multiple tensors. In addition, the method can suffer from overfitting when employing more than two core tensors, while it reduces accuracy when using just one tensor. As a result, subsequent advancements in tensor-based attention models have shown limited progress.

We introduce the Tensorized Attention Model (TAM) to efficiently integrate attention weights among multi-object relationships into the Transformer architecture by extending the existing Tensorized Transformer architecture (Ma *et al.*, 2019). TAM takes a multi-dimensional approach to representing complex real-world phenomena involving multiple object relationships. This enables us to consider "concurrent occurrences among objects", enhancing prediction accuracy. In this context, we introduce a third object type, 'semantic objects,' alongside the conventional query and memory objects. These semantic objects represent "shared and common semantic elements within query and memory objects". Our choice of the term 'semantic objects' emphasizes their pivotal role in capturing essential aspects of multi-object relationships. Semantic objects organically connect two different objects in the form of an independently shared axis. For example, in interconnected sentences, a common topic becomes a semantic object. Similarly, in Twitter discussions, specific locations can serve as semantic objects, just like shared conversational contexts in Reddit. Therefore, by introducing these semantic objects, we can effectively represent multi-object relationships.

TAM achieves these advancements by reimagining the current tensorized transformer framework through three key ideas. First, TAM employs Tucker decomposition based on object-dimensional sized matrices, as opposed to object-length sized matrices by Tensorized Transformer (Ma *et al.*, 2019). This adjustment ensures that the lengths of query, memory, and semantic vectors can vary while still enabling the calculation of multi-dimensional attention. TAM then enhances accuracy by implementing tensor decomposition that aggregates information from the memory and semantic components in a manner that aligns with the query length. Second, TAM initially computes multi-dimensional attentions among query, key (derived from memory), and semantics. It also leverages these multi-dimensional attentions to learn the transformation from the source (value from memory) to the target (query). Third, TAM employs an "iterative" process to average the attention computations obtained from Tucker decompositions. This approach ensures that the model consistently optimizes its memory usage for at most two sets of multi-dimensional attentions associated with two sets of core tensors, resulting in significant reductions in memory consumption compared to (Ma *et al.*, 2019), where memory allocation scales with the number of cores.

To showcase its effectiveness, we integrated TAM into the Transformer encoder and evaluated its performance in response selection tasks. By following the previous approach (Ma et al. (2019)), this paper focuses on measuring the impact of TAM's multi-dimensional attention within the encoder model although TAM could theoretically be applied to both encoder and decoder models. Our experiments followed two paths: training the TAM model from scratch and augmenting it with pre-trained Transformer models. The datasets employed for our evaluation were specifically designed to encompass key dimensions seen in natural dialogue, such as 'the context of the current dialogue', 'the entire history of a dialogue', and 'the topic under discussion in the dialogue'. When tested using the Reddit and TweetQA (Xiong *et al.*, 2019) datasets, which cover a wide range of topics, TAM consistently outperformed existing Transformer-based methods in terms of accuracy.

The paper's main contributions can be summarized in three key aspects: (1) We introduce TAM, a novel extension to the Transformer model that incorporates multi-dimensional attention mechanisms, enriching attention outputs by considering relationships across three or more distinct object types. (2) TAM innovates the tensorized transformer framework by employing Tucker decomposition for multi-object attention. It enhances accuracy through query-length-aligned tensor decomposition of key and value components. It also computes transformer outputs using multi-object attention in source-to-target transformations. It further reduces memory usage and overfitting through iterative averaging while maintaining accuracy. (3) We empirically validated TAM's effectiveness by integrating it into a Transformer encoder. Our experiments included training TAM from scratch and that with a pre-trained Transformer encoder. Evaluations on the Reddit and TweetQA datasets consistently demonstrated TAM's superior accuracy over existing Transformer-based techniques.

## 2 RELATED WORK

Several methods have explored tensor decomposition within transformer architectures to efficiently capture complex relationships, leading to parameter compression and improved learning efficiency (Bilgin *et al.*, 2022; Ma *et al.*, 2019; Shen *et al.*, 2019; Panahi *et al.*, 2021). For instance, (Shen *et al.*, 2019) introduces the Gated Multi-Head Attention (GMHA) module, which employs low-order matrices to represent large-scale weight matrices within a network layer. GMHA uses independent gates for each attention head to control attention values and reduce redundancy. However, despite these efforts to reduce parameters, experimental results indicate a decrease in accuracy.

(Ye *et al.*, 2020) employs low-rank block-term tensors to approximate weight matrix correlations in neural networks like CNNs and RNNs, enhancing representational capacity. (Hawkins *et al.*, 2022) presents an end-to-end framework for low-rank tensorized training, accommodating various low-rank tensor formats. Their efforts to reduce parameters, however, could lead to accuracy decreases in various scenarios. Importantly, these techniques have not been applied to Transformer attention mechanisms. Several studies propose efficient word embedding compression methods using tensor products (Hrinchuk *et al.*, 2020; Gan *et al.*, 2022). For instance, MorphTE (Gan *et al.*, 2022) combines morpheme vectors to represent word embeddings, integrating semantic and grammatical knowledge into the learning process. However, MorphTE specifically addresses word embedding compression and not the computation of multi-dimensional attention weights.

Actbert (Zhu and Yang, 2020) enhances encoding from three objects: action, regional, and linguistic features. It first blends action features from linguistic ones and guides action features from regional ones. It then computes source-target attentions from these blended or guided features to each target features. In contrast, TAM emphasizes the simultaneous observations of three distinct features.

Tensorized Transformer (Ma *et al.*, 2019) offers significant parameter reduction and performance enhancements compared to the standard Transformer. However, due to tensor decomposition limitations, it can only handle relationships between two types of objects, which necessitates equal query, key, and value lengths, restricting its use to self-attention tasks. This approach also directly derives the transformer output from attention weights among query, key, and value vectors, tailored for self-attention tasks, distinguishing it from scenarios requiring transformations between different object types. Furthermore, evaluation results show that increasing core tensors beyond two leads to higher memory overhead and overfitting, while a single core tensor may reduce accuracy.

## 3 PRELIMINARY

This section explains the Tucker decomposition and Multi-linear Attention, as our idea is based on these techniques. First we explain the notations of the paper. We use the Euler script letter $\mathcal{A}$ to denote a 3-order tensor, which can be thought of as a multi-array extending the concept of a matrix into three dimensions. Throughout this paper, we will use 3-order tensors for simplicity. However, it is worth noting that this approach can be extended to higher-dimensional tensors. In this specific context, the element in the 3-order tensor is denoted as $\mathcal{A}_{d_1,d_2,d_3}$. We use ":" as an index to fix certain dimensions in the tensor while representing the extraction of a face composed of the remaining dimensions; e.g. $\mathcal{A}_{d_1,d_2,:}$ signifies the extraction of a vector with the first and second dimensions fixed and the third dimension left open.

### 3.1 TUCKER DECOMPOSITION

Tucker decomposition (Tucker, 1966c) can model three-dimensional attention weights among the different types of objects. Given a 3-order tensor $\mathcal{A} \in \mathbb{R}^{d_1 \times d_2 \times d_3}$, we can decompose it into a core tensor $\mathcal{G}$ and three factor matrices $\mathbf{U}^{(1)}, \mathbf{U}^{(2)}, \mathbf{U}^{(3)}$ by using Tucker decomposition. Here $\mathcal{G} \in \mathbb{R}^{r_1 \times r_2 \times r_3}$, $\mathbf{U}^{(1)} \in \mathbb{R}^{d_1 \times r_1}$, $\mathbf{U}^{(2)} \in \mathbb{R}^{d_2 \times r_2}$, and $\mathbf{U}^{(3)} \in \mathbb{R}^{d_3 \times r_3}$. Tucker decomposition can be formulated as follows where $\times_k$ denotes the tensor-matrix product along the $k$-th mode: $\mathcal{A} = \mathcal{G} \times_1 \mathbf{U}^{(1)} \times_2 \mathbf{U}^{(2)} \times_3 \mathbf{U}^{(3)}$.

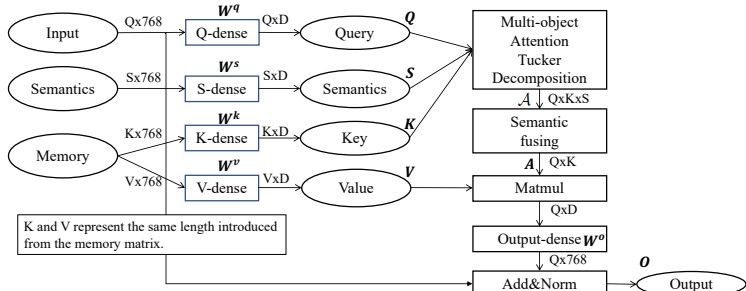

Figure 1: The attention architecture of the TAM.

## 3.2 MULTI-LINEAR ATTENTION BY BLOCK-TERM TENSOR DECOMPOSITION

This paper bases our ideas on a specialized attention mechanism, referred to as Multi-Linear Attention, which is built for the multi-head attention focusing on self-attention mechanism (Ma *et al.*, 2019). First, it assumes that the query, key, and value can be mapped into sets of three orthogonal basis vectors. Each of these factor matrices (i.e. $\mathbf{Q}$, $\mathbf{K}$, and $\mathbf{V}$) has a dimension of $N \times d$, where $N$ is the sequence length and $d$ is the dimensionality of the matrix. It next initializes a trainable 3-order diagonal tensor $\mathcal{G}$ of rank $R$. The single-head attention mechanism is then defined as follows:

$$\text{AttenTD}(\mathcal{G}; \mathbf{Q}, \mathbf{K}, \mathbf{V}) = \sum_{q=1}^{Q} \sum_{k=1}^{K} \sum_{v=1}^{V} \mathcal{G}_{q,k,v} \left( \mathbf{Q}_{q,:} \odot \mathbf{K}_{k,:} \odot \mathbf{V}_{v,:} \right). \quad (1)$$

AttenTD is an attention weight tensor that stores attention weights among the query, key, and value as its elements. Here, $\mathcal{G}$ serves as the core tensor, while $q$, $k$, and $v$ are its indices. $Q$, $K$, and $V$ represent the lengths of the query, key, and value, respectively, while The symbol $\odot$ represents the outer product. $\mathbf{Q}_{q,:}$, $\mathbf{K}_{k,:}$, and $\mathbf{V}_{v,:}$ are column vectors extracted from $\mathbf{Q}$, $\mathbf{K}$, and $\mathbf{V}$ respectively. In practice, it assumes that $Q = K = V = R$. This core tensor $\mathcal{G}$ is initialized such that:

$$\mathcal{G}_{q,k,v} = \begin{cases} 0 & \text{if } q \neq k \text{ or } q \neq v \text{ or } k \neq v \\ g_r = \text{rand}(0,1) \text{ s.t. } \sum_{r=1}^{R} g_r = 1 & \text{if } q = k = v = r \end{cases} \quad (2)$$

The tensor $\mathcal{G}$ forms the weight vector $\mathbf{g}$ with trainable elements $g_r$, computed along its diagonal using the softmax function.

After that, it applies the Block-Term tensor decomposition to build multi-head mechanism, named as Multi-linear attention, which can be formulated as follows:

$$\text{MultiLinear}(\mathcal{G}; \mathbf{Q}, \mathbf{K}, \mathbf{V}) = \text{SplitConcat} \left( \frac{1}{H} \times (T_1 + \ldots + T_h + \ldots + T_H) \right) \mathbf{W}^O \quad \text{s.t.} \quad (3)$$

$$T_h = \text{AttenTD}(\mathcal{G}_h; \mathbf{Q}\mathbf{W}^q, \mathbf{K}\mathbf{W}^k, \mathbf{V}\mathbf{W}^v)$$

MultiLinear$(\mathcal{G}; \mathbf{Q}, \mathbf{K}, \mathbf{V})$ represents the attention output matrix which is computed directly from the attention weight tensors, AttenTDs, by applying a linear function. The core tensor $\mathcal{G}_h$ is a diagonal tensor, and the number of parameters in $\mathcal{G}_h$ is equal to the rank of the core tensor, $h \in \{1, \ldots, H\}$. $\mathcal{G}$ is the set of the core tensors. SplitConcat$(\cdot)$ is a function which achieves the concatenation after splitting for a 3-order tensor. $\mathbf{W}^O$ is the parameter matrix which is a fully connected layer and is correlated to the output of Multi-linear attention. $\mathbf{W}^q$, $\mathbf{W}^k$, and $\mathbf{W}^v$ are parameter matrices that are learned to adjust the dimension size of $\mathbf{Q}$, $\mathbf{K}$, $\mathbf{V}$, respectively, and they are shared when constructing multi-core tensors in Multi-linear attention.

In the evaluation conducted by (Ma *et al.*, 2019), the model achieves its highest prediction accuracy when the number of attention heads $H$ is set to two while increasing $H$ beyond this value leads to overfitting during model training.

## 4 METHOD

This section explains the details of our new multi-object attention model, TAM.

## 4.1 BASIC IDEA AND OUR ATTENTION ARCHITECTURE

First, we introduce our idea: a novel multi-object attention mechanism that captures real-world relationships. In conventional self-attention mechanisms, a query ($\mathbf{Q} \in \mathbb{R}^{Q \times D}$), a key from a memory ($\mathbf{K} \in \mathbb{R}^{K \times D}$), and a value from a memory ($\mathbf{V} \in \mathbb{R}^{V \times D}$) are employed where $D$ represents the dimension size. In standard Self-Attention, $\mathbf{Q}$, $\mathbf{K}$, and $\mathbf{V}$ are identical, while in Source-Target Attention, $\mathbf{Q}$ and $\mathbf{K}$ come from different sources, making them different. In our approach, we introduce an additional component, "semantics" ($\mathbf{S} \in \mathbb{R}^{S \times D}$), which is distinct from the existing $\mathbf{Q}$, $\mathbf{K}$, and $\mathbf{V}$ components to represent multi-object relationships. Specifically, our Tensorized Attention framework operates such that $\mathbf{Q}$, $\mathbf{K}$, and $\mathbf{S}$ are all distinct from each other, while $\mathbf{K}$ and $\mathbf{V}$ remain the same since they are from the same source, memory. This innovation enables our attention mechanism to capture complex dependencies and interactions more effectively by leveraging the semantic information encapsulated in the "semantics" ($\mathbf{S}$) component. As a result, our mechanism can more accurately model natural relationships in the real-world context.

The architecture of TAM based on the above idea is depicted in Fig. 1. Unlike conventional attention models, TAM takes not only query and memory but also semantics as inputs. Here, each of these inputs first passes through a dense network, where $\mathbf{W}^q$ represents the weight matrix for queries, $\mathbf{W}^s$ for semantics, $\mathbf{W}^k$ for keys, and $\mathbf{W}^v$ for values, respectively. Following this, the model calculates three-dimensional attention tensor $\mathcal{A}$ among the query matrix $\mathbf{Q}$, semantics matrix $\mathbf{S}$, and key matrix $\mathbf{K}$ by Multi-object Attention with Tucker Decomposition. To incorporate these three-dimensional attention weight matrices into the value matrix $\mathbf{V}$, we convert $\mathcal{A}$ into a two-dimensional matrix $\mathbf{A}$ through semantic fusion, which involves summarizing the $q \times k$ matrices along the semantic direction. Subsequently, we perform Matmul (matrix multiplication) between this 2D matrix $\mathbf{A}$ and the value matrix $\mathbf{V}$. Afterward, we apply Add&Norm, i.e., addition and normalization, (Vaswani *et al.*, 2017) through a dense network between this updated value and the original query. Ultimately, this process yields the output $\mathbf{O}$ of the Transformer encoder when TAM is integrated. In our experiments, we replace the attention layer of the BERT implementation with TAM.

## 4.2 MULTI-OBJECT ATTENTION TUCKER DECOMPOSITION & SEMANTIC FUSING

The limitation of (Ma *et al.*, 2019) lies in the fact that the length of matrices for $\mathbf{Q}$, $\mathbf{K}$, and $\mathbf{V}$ assumes to be the same. However, real-world multi-dimensional object relationships naturally involve objects of varying lengths. So, we revise Eq. (1) into the following Eq. (4) by $D$-length column vectors $\mathbf{Q}_{q,:}$, $\mathbf{K}_{k,:}$, and $\mathbf{S}_{s,:}$, which are extracted from $\mathbf{Q}$, $\mathbf{K}$, and $\mathbf{S}$ respectively; the length of column vectors is consistently the dimension size, making them all of the same length, denoted as $D$:

$$\mathcal{A}_{q,k,s} = \sum_{i=1}^{D} \sum_{j=1}^{D} \sum_{l=1}^{D} \mathcal{G}_{q,j,l} \cdot (\mathbf{Q}_{:,i} \odot \mathbf{K}_{k,:} \odot \mathbf{S}_{s,:})_{q,:,:,j,:,l} \tag{4}$$

Here, the core tensor size is set to $\mathbb{R}^{Q \times D \times D}$, with length $Q$ in $\mathbf{Q}$ retained on the core tensor side, enabling alignment centered around $Q$ for its relationships with $\mathbf{K}$ and $\mathbf{S}$. This is the reason we select the $Q$-length row vector $\mathbf{Q}_{:,i}$ when computing the outer product among three matrices and performing summation along the $D$ dimension with respect to $\mathbf{Q}$. In this equation, $(\mathbf{Q}_{:,i} \odot \mathbf{K}_{k,:} \odot \mathbf{S}_{s,:})_{q,:,:,j,:,l}$ is the outer product where first, fourth, and sixth dimensions are activated in $(\mathbf{Q}_{:,i} \odot \mathbf{K}_{k,:} \odot \mathbf{S}_{s,:})$. In Eq. (4), "$\cdot$" represents dot product among two tensors. The core tensor $\mathcal{G}$ has a $D$-length trainable weight vector $\mathbf{g}$ primarily on its diagonal and is initialized as below equation.

$$\mathcal{G}_{q,k,s} = \begin{cases} 0 & \text{if } q \neq k \text{ or } q \neq s \text{ or } k \neq s \text{ or } D < q \\ g_d = \text{rand}(0,1) \text{ s.t. } \sum_{d=1}^{D} g_d = 1 & \text{if } q = k = s = d \end{cases} \tag{5}$$

Each entry in $\mathbf{g}$ can be represented as $g_d$ for $d \in \{1, \ldots, D\}$. If $D < Q$, then the element where $D < q$ becomes zero. If $D \geq Q$, the length of $\mathbf{g}$ becomes equal to the length of Q, i.e. $D = Q$.

The previous study (Ma *et al.*, 2019) hires a method known as "split&concat" that was used to convert the multi-dimensional attention into a two-dimensional form. This method splits the three-dimensional attention obtained from Attention Tucker Decomposition along the $V$-axis and then concatenates them to create 2D-matrix whose size is $\mathbb{R}^{Q \times (S \cdot K)}$. After that, this 2D-matrix was transformed into the transformer output whose size is $\mathbb{R}^{Q \times D}$ through a Linear layer. However, this

method diverges from the original strength of the Transformer especially when handling transformations from a source object to a different type of target object. This is because it cannot effectively learn such transformations although the conventional Transformer excels at transforming $\mathbf{V}$ using attention output and then seamlessly integrating it with $\mathbf{Q}$.

To address these limitations, TAM collapses the $S$-axis of the multi-dimensional attention tensor to effectively convert it into a two-dimensional matrix. Specifically, Eq. (4) can be re-formulated as:

$$\mathbf{A}_{q,k} = \sum_{s=1}^{S} \left[ \sum_{i=1}^{D} \sum_{j=1}^{D} \sum_{l=1}^{D} \mathcal{G}_{q,j,l} \cdot (\mathbf{Q}_{:,i} \odot \mathbf{K}_{k,:} \odot \mathbf{S}_{s,:})_{q,:,:,j,:,l} \right]_{:,:,s} . \quad (6)$$

By doing so, TAM calculates 2D attention between $\mathbf{Q}$ and $\mathbf{K}$ based on a semantically rich 3D attention tensor. Moreover, by performing an inner product with $\mathbf{V}$ as below, the model is able to reflect the transformation by the attention mechanism within $\mathbf{V}$ itself. Finally, an 'Add&Norm' operation is conducted with $\mathbf{Q}$ to generate the final output $\mathbf{O}$ as: $\mathbf{O} = \text{Add\&Norm}(\mathbf{A}_{q,k}\mathbf{V}, \mathbf{Q})$.

## 4.3 Multi-core attention computations

TAM also hires multiple core tensors to compute multi-dimensional attentions by Tucker decompositions and averages them to build final multi-dimensional attention weights. Unlike the approach presented in (Ma *et al.*, 2019), TAM significantly improves memory efficiency when predicting multi-dimensional attentions. It achieves this by dynamically summing three-dimensional attention tensors on-the-fly; we can eliminate the need to calculate and store each tensor separately before summing them and subsequently collapsing them along the S-axis during computation. This approach enhances memory utilization. Furthermore, as previously mentioned, TAM overcomes the $Q = K = L$ constraint by prioritizing tensor decomposition centered around D. These capabilities enhance accuracy, even when utilizing a single core, and consistently deliver robust performance without overfitting or memory concerns, even when configured with three or more cores. The equation is iteratedly computed where $1 \le n \le (N-1)$ as follows:

$$\mathbf{A}_{q,k}^{n+1} = \mathbf{A}_{q,k}^{n} + \Delta^{n} \quad \text{s.t.} \quad \Delta^{n} = \sum_{s=1}^{S} \left[ \sum_{i=1}^{D} \sum_{j=1}^{D} \sum_{l=1}^{D} \mathcal{G}_{q,j,l}^{n} \cdot (\mathbf{Q}_{:,i} \odot \mathbf{K}_{k,:} \odot \mathbf{S}_{s,:})_{q,:,:,j,:,l} \right]_{:,:,s} \quad (7)$$

where $n$ is the index of the $N$ number of core tensors and $\mathcal{G}_{q,j,l}^{n}$ is $n$-th core tensor. After each iteration, the previous $\mathbf{A}_{q,k}^{n}$ is discarded to save memory. Finally, $\mathbf{A}_{q,k}$ is computed as: $\frac{1}{N}\mathbf{A}_{q,k}^{N}$.

## 4.4 Combining the Pre-trained language model

While TAM is powerful for its ability to naturally handle real-world relationships in multi-dimensions, it can also be combined with pre-trained language models based on conventional 2D-attention mechanisms such as BERT-base that are widely used today. We achieve this integration by feeding the output from the pre-trained language model, along with corresponding semantic information, into the TAM as inputs. By doing so, TAM can also serve with the pre-train model, extending its utility across various applications.

## 4.5 Training the model

TAM pretrains the Masked Language Model (MLM) and Next Segment Prediction (NSP) using input, memory, and semantics information, as illustrated in Fig. 1. The model is optimized using the sum of cross-entropy losses for MLM and NSP, following the BERT training. In fine-tuning for response selection, as in our evaluation, TAM utilizes the hidden vector $\mathbf{O}_{cls}$ extracted from the first cls token in $\mathbf{O}$. It then computes the matching score between the current utterance context and the response using a single-layer neural network, denoted as $\sigma(\mathbf{W} \cdot \mathbf{O}_{cls})$, where $\mathbf{W}$ represents a trainable parameter. The model weights are then updated using the cross-entropy loss function.

# 5 Evaluation

This section evaluates TAM's performance in response selection tasks.

Table 1: Statistics of the datasets.

| | NFL | | Politics | |
|---|---|---|---|---|
| | Training | Test | Training | Test |
| Number of dialogues | 230,060 | 13,765 | 290,020 | 19,040 |
| Avg. turns per dialogue | 4.2 | 4.2 | 4.8 | 4.9 |
| Avg. words per dialogue | 56.3 | 57.6 | 81.1 | 81.5 |

Table 2: Comparison among the methods when learning from scratch.

| Method | NFL | | | | Politics | | | |
|---|---|---|---|---|---|---|---|---|
| Metrics | $R_{10}@1$ | $R_{10}@2$ | $R_{10}@5$ | #params | $R_{10}@1$ | $R_{10}@2$ | $R_{10}@5$ | #params |
| BERT(CR) | 34.08 | 51.28 | 81.56 | 146,167,049 | 37.62 | 55.46 | 83.37 | 146,167,269 |
| BERT(SCR) | 35.76 | 53.39 | 82.44 | 146,167,049 | 37.09 | 54.43 | 82.75 | 146,167,269 |
| *Tensorized Transformer(CR)* | 35.19 | 52.39 | 81.97 | 160,170,977 | 34.24 | 51.37 | 80.15 | 223,873,677 |
| *Tensorized Transformer(SCR)* | 32.05 | 49.39 | 79.32 | 169,342,001 | 36.37 | 53.54 | 81.73 | 315,644,781 |
| *TAM* | ***45.61*** | ***61.86*** | ***85.57*** | 156,228,041 | ***51.51*** | ***65.92*** | ***87.65*** | 155,047,461 |

## 5.1 DATASET

We used the Reddit dataset to evaluate TAM in the task of the response selection. Two datasets, "NFL" and "Politics", were compiled by sampling posts from their respective communities between September 2018 and February 2019. The dataset is accessible through BigQuery (https://bigquery.cloud.google.com/dataset/fh-bigquery). We focused on active speakers and paired their comments with responses, dividing the data into training and test datasets. Those comments and their responses, which are being discussed among users, form "a threaded dialogue". See Table 1 for dataset statistics. The response selection task aims to identify the last utterance (response) in a dialogue based on the utterance context leading up to that response. In our evaluation, we focused on the following three-object relationships as they encompass key dimensions observed in natural dialogue: (1) 'The entire history of a dialogue': This is derived from the word embedding stream for the current dialogue and can serve as **Q**. We utilized the BERT-base tokenizer. (2) 'The context of the current dialogue': Created by applying GRU (Cho *et al.*, 2014) to the word embedding streams in both the utterance context and the response. The hidden embedding streams for the utterance context and its response are concatenated, ensuring an unbiased representation of context by partitioning the GRU streams before concatenation. This is represented as either **K** or **V**. (3) 'The topic under discussion in the dialogue': Obtained from the word embedding stream for the subject title of the Reddit dialogue and represented as **S**. We set the dialogue length to 70 and title length to 35 for the NFL dataset, and 180 for dialogue length and 60 for title length for the Politics dataset. These settings cover 95% of the dialogues and titles in both datasets. We also conducted evaluations on the TweetQA dataset. For the results pertaining to the TweetQA dataset, please refer to the Appendix.

## 5.2 COMPARED METHODS

We integrate *TAM* into the BERT implementation, as explained in Section 4.1, by replacing its attention layer. By directly comparing *TAM* with the BERT model before *TAM* integration, we can effectively isolate and analyze the differences in multi-dimensional attention. This approach provides a clear and comprehensive validation of *TAM*'s effectiveness in contrast to the standard Transformer encoder. To ensure a fair evaluation, we use two variants of BERT: one that inputs 'the entire history of a dialogue,' as query, key, and value denoted as *BERT(CR)*, and another that inputs a concatenation of 'the topic under discussion in the dialogue' and 'the entire history of a dialogue,' denoted as *BERT(SCR)*. Additionally, we introduce two variants, *Tensorized Transformer(CR)* and *Tensorized Transformer(SCR)*, for comparison, following a similar configuration as BERT. All models utilize 12 transformer encoder layers.

Furthermore, in our experiments with pre-trained models (see Section 4.4), we followed three approaches: (1) we created a hybrid model by integrating the outputs of 12 layers of $BERT^P(SCR)$ into the TAM model, denoted as $BERT^P(SCR)$-*TAM*. (2) we used the output from *BERT(CR)* or *BERT(SCR)* as input for *BERT(CR)* or *BERT(SCR)*, labeled as $BERT^P(CR)$-*BERT(CR)* or $BERT^P(SCR)$-*BERT(SCR)*. (3) we also compared our model with a single 12 layers of $BERT^P(SCR)$. Here, the term "$BERT^P$" refers to the pre-trained BERT-base model (12 layers), while the terms "BERT" or "TAM" indicate unpretrained models with 4 layers. The choice of 4 layers was made due to its superior performance and computational efficiency compared to the 12-layer model.

Table 3: Comparison when leveraged a pretrained language model for initialization.

| Method | NFL | | | Politics | | |
|---|---|---|---|---|---|---|
| Metrics | $R_{10}@1$ | $R_{10}@2$ | $R_{10}@5$ | $R_{10}@1$ | $R_{10}@2$ | $R_{10}@5$ |
| $BERT^P(CR)$-BERT(CR) | 65.70 | 78.50 | 93.80 | 70.11 | 81.63 | 94.73 |
| $BERT^P(SCR)$ | 66.80 | 79.27 | 94.24 | 70.95 | 82.33 | 95.06 |
| $BERT^P(SCR)$-BERT(SCR) | 68.04 | 80.36 | 94.78 | 71.73 | 83.01 | 95.19 |
| $BERT^P(SCR)$-TAM | **68.54** | **80.94** | **94.89** | **72.07** | **83.32** | **95.48** |

Table 4: Ablation study when learning from scratch.

| Method | NFL | | | Politics | | |
|---|---|---|---|---|---|---|
| Metrics | $R_{10}@1$ | $R_{10}@2$ | $R_{10}@5$ | $R_{10}@1$ | $R_{10}@2$ | $R_{10}@5$ |
| TAM | 45.61 | 61.86 | 85.57 | 51.51 | 65.92 | 87.65 |
| w/o Semantic fusing | 42.74 | 59.41 | 84.21 | 47.97 | 63.81 | 86.56 |
| w/o query aligned | 38.01 | 55.05 | 81.98 | 32.45 | 50.02 | 79.13 |

## 5.3 METRICS

Our evaluation metric is $Recall_{10}@k$ ($R_{10}@k$) (Qian *et al.*, 2021; Han *et al.*, 2021). Given ten responses, the evaluation measures if the relevant response is ranked among the top $k$ candidates. This metric hires 9 responses randomly sampled from the test dataset as negatives. This takes into account the importance of both top-1 and lower-ranked predictions in diverse applications. The word embeddings had an embedding size of 768 following (Devlin *et al.*, 2019). The learning rate was set to $1 \times 10^{-5}$. We use the AdamW optimizer (Loshchilov and Hutter, 2019) with beta values of 0.9 and 0.999, respectively, and an epsilon of $1.0 \times 10^{-8}$. Pre-training was performed for 20 epochs, and fine-tuning was done for 15 epochs on both datasets for all methods when learning from scratch. In experiments with pre-trained BERT-base models, pre-training for $BERT^P(SCR)$-TAM was performed for 6 epochs, and fine-tuning was done for 6 epochs on both datasets. Methods, except for $BERT^P(SCR)$-TAM, required 15 epochs for fine-tuning on both datasets. These epochs were sufficient for model convergence. The *Tensorized Transformer* utilizes two cores for optimal performance, as a single core tends to underperform, and using three or more cores is hindered by memory constraints. For *TAM*, we configured the core count to be three for the NFL dataset and twenty for the Politics dataset. We confirmed the reproducibility with five random seeds for accuracy comparisons. We set the dimension size of $\mathbf{Q}$, $\mathbf{K}$, $\mathbf{V}$, and $\mathbf{S}$ to 192 for the NFL dataset and 160 for the Politics dataset. The batch size was 96 for both datasets for all methods. The hardware used for these tests was an NVIDIA A100 GPU with 80GB of memory.

## 5.4 RESULT

Table 2 presents results from training the model from scratch, without using any pre-trained language models. Firstly, *BERT(SCR)* demonstrates higher accuracy than *BERT(CR)* on the NFL dataset, while the reverse holds true for the Politics dataset. These results suggest that semantic information may be effective in improving accuracy, but the current attention model struggles to fully leverage the potential benefits of different object types. Secondly, both *Tensorized Transformer(CR)* and *Tensorized Transformer(SCR)* exhibit only modest improvements or slight decreases compared to BERT across both datasets. This aligns with expectations since the primary aim of *Tensorized Transformer* is to enhance memory efficiency in self-attention for single objects, limiting its ability to achieve significant accuracy improvements in multi-object scenarios. Finally, *TAM* achieves remarkably high accuracy on both datasets when trained from scratch. This emphasizes the importance of employing multi-object tensors to naturally represent real-world relationships. TAM's capability to aggregate information from various objects around $\mathbf{Q}$ and facilitate transformations between source and target objects significantly improves prediction accuracy. As for model parameters, *TAM* shows a marginal increase compared to *BERT*, encompassing additional weights in dense networks and core tensor weights. Conversely, *Tensorized Transformer* sees a significant rise in parameters, particularly in the Politics dataset, attributed to the length-dependent $\mathbf{W}^o$ values associated with $K \times V$.

Next, Table 3 presents the results when leveraging a pretrained language model for initialization. $BERT^P(SCR)$-BERT(SCR) outperforms $BERT^P(CR)$-BERT(CR) and $BERT^P(SCR)$ in terms of accuracy. Additionally, $BERT^P(SCR)$-TAM achieves higher accuracy than $BERT^P(SCR)$-BERT(SCR). These findings indicate that incorporating semantic information to train multi-object attention is also beneficial when combining TAM with a pre-trained traditional language model.

Table 5: Performance of TAM (*BERT(SCR)-TAM*) against various batch sizes and dimension sizes.

| | | NFL | | | | | Politics | | |
|---|---|---|---|---|---|---|---|---|---|
| batch | dim | $R_{10}@1$ | $R_{10}@2$ | $R_{10}@5$ | batch | dim | $R_{10}@1$ | $R_{10}@2$ | $R_{10}@5$ |
| 96 | 160 | 68.52 | 81.04 | 95.05 | 96 | 128 | 72.19 | 83.53 | 95.50 |
| 96 | 192 | 68.54 | 80.94 | 94.89 | 96 | 160 | 72.07 | 83.32 | 95.48 |
| 96 | 256 | 68.51 | 80.83 | 94.94 | 96 | 192 | 71.95 | 83.33 | 95.41 |
| 64 | 160 | 68.36 | 80.97 | 94.85 | 64 | 128 | 72.09 | 83.71 | 95.42 |
| 64 | 192 | 68.36 | 81.22 | 95.04 | 64 | 160 | 72.11 | 83.49 | 95.37 |
| 64 | 256 | 68.54 | 80.91 | 94.82 | 64 | 192 | 72.06 | 83.23 | 95.41 |

Table 6: $R_{10}@1$ with variations in the number of cores.

| | NFL | | | | Politics | | | |
|---|---|---|---|---|---|---|---|---|
| Number of Cores | 1 | 2 | 3 | 20 | 1 | 2 | 3 | 20 |
| *TAM* | 44.98 | 45.39 | 45.61 | 43.04 | 50.86 | 49.88 | 49.84 | 51.51 |

Qualitative evaluation, including visualized examples of attention weights computed for both $TAM$ and $BERT(SCR)$, can be found in the appendix A.1 for further details.

## 5.5 ABLATION STUDY

For the ablation study, we consider the "*w/o Semantic Fusing*" method, employing the same input configuration as *TAM*: using the query as the dialogue history, the key and value as the current utterance context, and the semantic as the dialogue title. However, *w/o Semantic Fusing* employs *split&concat* to compute the transformer output directly from the attention weight tensor through q-k-s attention tensor. We also consider "*w/o Query Aligned*", which is another variant that shares the TAM method but with a core tensor size of $\mathbb{R}^{D \times D \times D}$, not $\mathbb{R}^{Q \times D \times D}$ (refer to Eq. (4)). This implies a lack of alignment centered around $Q$.

Table 4 presents the results of the ablation study. *w/o Semantic Fusing* falls short compared to *TAM* because it does not learn the transformations from source objects to different target objects using attention weight tensors. *w/o Query Aligned* also exhibits lower accuracy than *TAM* due to the latter's ability to aggregate information from other objects around **Q** through tensor decomposition. This aligns with the transformer's process of updating **Q**, providing an explanation for the observed performance gap. Table 5 displays the performance of *TAM* as we vary both the batch size and dimension size. When considering this table alongside Table 2, it becomes evident that the accuracy remains consistently high, regardless of batch size or dimension size, outperforming other methods and demonstrating stability. This stability stems from the model's flexibility, relying on the low-rank $D$ in the core tensor for calculations, rather than being limited by Query or Memory lengths.

## 5.6 TENSOR DECOMPOSITION CORES AND TAM RESULTS

The results of TAM, with variations in the number of cores used in tensor decompositions, are presented in Table 6. Our approach employs a memory-efficient Tucker decomposition with multiple cores as explained in Section 4.3. As a result, our approach can efficiently harness as many as twenty cores in tensor decompositions, whereas the *Tensorized Transformer* in our evaluation setting only utilizes up to two cores as mentioned in Section 5.3. We observed that the optimal number of cores depends on the dataset. For instance, *TAM* achieves higher accuracy with three cores for the NFL dataset and twenty cores for the Politics dataset. Furthermore, *TAM* is much more efficient than *Tensorized Transformer*, as present in Table 2. Despite increasing the number of cores, the parameter count barely increases, thanks to the memory-efficient Tucker decomposition.

## 6 CONCLUSION

This paper introduces the Tensorized Attention Model (TAM), which employs Tucker decompositions with multi-core tensors for memory-efficient computation of attention weights across diverse object types, seamlessly integrating them into the Transformer encoder. The evaluation on the Reddit dataset demonstrates that TAM outperforms the standard Transformer and previous tensorized transformers in terms of accuracy. We intend to employ TAM in future research for generative models based on Transformer decoders, particularly on much larger datasets.

## REPRODUCIBILITY STATEMENT

Our methodology is currently awaiting patent approval. Once the patent is granted, we will release the code to the public with permission from our affiliated organization. For our experiments, we utilized an openly available Reddit dataset, as explained in Section 5.1. Creating datasets for experiments can be achieved as long as there are triples of titles, comments, and replies. We have ensured reproducibility through experiments conducted with five random seeds, as mentioned in Section 5.3. The code for the tensorized transformer, the foundation of this paper (Ma *et al.*, 2019), is publicly accessible on GitHub (https://github.com/szhangtju/The-compression-of-Transformer). Given the simplicity of the equations introduced in our method in Section 4, reproducing our results is straightforward, enabling readers to easily replicate our experiments.

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

# A APPENDIX

## A.1 MEANINGFUL RESULTS

We provide visualized examples of attention weights computed for both $TAM$ and $BERT(SCR)$ in Fig. 2. The colors in the visualization represent the level of attention weight, with yellow indicating high attention, green representing neutrality, and blue suggesting low attention. In *TAM*, we represent utterance contexts on the x-axis and responses on the y-axis. In *BERT(SCR)*, we depict semantics and utterance contexts on the x-axis. We have selectively labeled certain words on the X and Y axes for improved readability. In this example, semantics is "[Highlight] Marcus Peters intercepts Mahomes with 1:13 remaining", utterance context is "He's a rookie. Sports went up +'s be turnovers.", and reply is "Can't believe donovan mitchell is going to lose roty 2 years in a row".

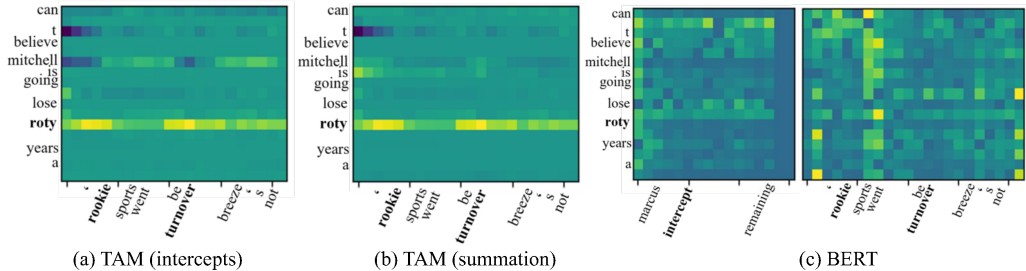

|     (a) TAM (intercepts)     |     (b) TAM (summation)     |     (c) BERT     |

Figure 2: Visualization examples of attention weights for $TAM$ and $BERT$.

Table 7: Comparison among the methods when learning from scratch.

| Method | TweetQA | | | |
|---|---|---|---|---|
| Metrics | $R_{10}@1$ | $R_{10}@2$ | $R_{10}@5$ | #params |
| BERT(CR) | 19.98 | 33.79 | 63.72 | 146,167,005 |
| BERT(SCR) | 23.39 | 37.48 | 69.52 | 146,167,005 |
| *Tensorized Transformer(CR)* | 14.18 | 26.80 | 58.38 | 157,474,437 |
| *Tensorized Transformer(SCR)* | 19.61 | 36.19 | 72.01 | 158,952,501 |
| *TAM* | ***47.15*** | ***56.63*** | ***75.14*** | 155,047,197 |

Fig. 2-(a) displays the sliced $q \times k$ matrix corresponding to the semantic token "intercept" by TAM. Additionally, Fig. 2-(b) presents the summation of all $q \times k$ matrices for semantic tokens (refer to Eq. (6)) by TAM. In Fig. 2-(a), $TAM$ effectively captures relationships among the words "intercepts" in semantics, "rookie" and "turnovers" in the utterance context, and "roty (rookie of the year)" in the response. These relationships are crucial, as they pertain to impressive interceptions and turnover plays that are fundamental for a rookie. In Fig. 2-(b), $TAM$ retains these relationships even after summing the weights across the semantic tokens. In contrast, $BERT(SCR)$ fails to identify such relationships among the three different object types, as demonstrated in Fig. 2-(c).

In Reddit, casual language and fragmented sentences are often used, making it challenging to extract context from text alone. However, TAM effectively manages to capture the relationships between semantics, context, and response. In contrast, approaches like BERT that simply prepend semantics to the beginning of the text reduce these multi-dimensional relationships to two dimensions. Consequently, they may fail to accurately capture simultaneous relationships among semantics, context, and response. In the given example, they might not effectively highlight the central topic of Mahomes' rookie season and the critical attributes of rookies in the NFL.

## A.2 EVALUATION ON QA DATASET

We conducted the evaluation on the Tweet QA dataset (Xiong *et al.*, 2019). It is a large-scale dataset designed for automated question answering, permitting abstractive responses over social media content. It specifically emphasizes meaningful tweets employed by journalists in the creation of news articles. It consists of question-passage-answer triples. The passages are tweets that have been used by journalists in news articles, implying that such tweets contain useful information. It includes 10,692 training triples and 1,979 test triples. The average question length is 6.95, and the average answer length is 2.45.

We prepared the following three-object relationships as we did in Section 5.1: "The entire topic of a passage", which is derived from the word embedding stream for the passage and can serve as $\mathbf{Q}$. "The context of the passage-answer pair", which is created by applying GRU (Cho *et al.*, 2014) to the word embedding streams in both the passage and the answer. This is represented as either $\mathbf{K}$ or $\mathbf{V}$. "The question assigned for the passage", which is obtained from the word embedding stream for the question and represented as $\mathbf{S}$.

We set the passage-answer length to 48 and the question length to 15 for this dataset. These settings cover 95% of the passage-answer pairs and questions in both datasets. Pre-training was performed for 100 epochs, and fine-tuning was done for 20 epochs on both datasets for all methods since all methods converge until the above epochs. We set the dimension size $D$ of $\mathbf{Q}$, $\mathbf{K}$, $\mathbf{V}$, and $\mathbf{S}$ to 160 for the Tweet QA dataset. The other parameters were set to the same values as in the Reddit dataset, as detailed in Section 5.3.

Table 8: Ablation study when learning from scratch.

| Method | TweetQA | | |
|---|---|---|---|
| Metrics | $R_{10}@1$ | $R_{10}@2$ | $R_{10}@5$ |
| *TAM* | 47.15 | 56.63 | 75.14 |
| *w/o Semantic fusing* | 40.61 | 54.14 | 76.61 |
| *w/o query aligned* | 14.73 | 26.52 | 61.79 |

Table 9: Evaluating accuracy and parameter size with varied number of layers in *BERT(SCR)*.

| Method | NFL | | | | Politics | | | |
|---|---|---|---|---|---|---|---|---|
| Num of layers | $R_{10}@1$ | $R_{10}@2$ | $R_{10}@5$ | #params | $R_{10}@1$ | $R_{10}@2$ | $R_{10}@5$ | #params |
| 12 | 35.30 | 53.09 | 82.90 | 146,167,049 | 40.32 | 52.75 | 83.65 | 146,167,269 |
| 15 | 37.55 | 55.60 | 83.33 | 167,430,665 | 43.02 | 59.76 | 85.29 | 167,430,885 |
| 18 | 37.59 | 55.39 | 84.08 | 188,694,281 | 43.33 | 60.24 | 85.26 | 188,694,501 |
| 21 | 37.04 | 54.83 | 83.47 | 209,957,897 | 44.44 | 61.25 | 85.96 | 209,958,117 |

The results when learning from scratch are present in Table 7. The tendency of the results is almost the same as those of the Reddit dataset. Interestingly, *TAM* achieves much higher accuracy than other methods, including *BERT(SCR)* and *Tensorized Transformer(SCR)*, compared to the improvements observed in the Reddit dataset. We note that *BERT(SCR)* and *Tensorized Transformer(SCR)* use the same inputs (e.g., questions, passages, and answers). They concatenate questions with passage-answer pairs in a similar way as done in the Reddit dataset in Section 5.2. We attribute this significant improvement to the stronger associations among questions and passage-answer pairs, in contrast to the relationships observed among titles and utterances within the Reddit dataset. We also consider that effective utilization of multi-objective relationships in a smaller-sized dataset like TweetQA is essential, as it maximizes the use of the observed limited dataset, in comparison to the Reddit dataset. *TAM* can effectively learn the co-occurrences of multi-object relationships such as those among tokens in **Q**, **K**, and **S** from the restricted amount of observations.

Table 8 demonstrates the effectiveness of our semantic fusing and query-aligned approaches in improving accuracy. Particularly, the results indicate that the query-aligned approach is successful in enhancing accuracy by focusing on the alignment centered around $Q$ for its relationships with **K** and **S** in learning the core tensor. This is believed to occur because the explicit token embedding of the Query can incorporate the latent semantics derived from the correlation between $q$, $k$, and $v$ in an explicit manner.

## A.3 PARAMETER SIZE ANALYSIS: TAM VS. BERT IN LAYER VARIATION ABLATION STUDY

We conducted an ablation study by varying the number of layers in *BERT(SCR)* and *TAM* using Reddit dataset. The results are summarized in Tables 9 and 10. For *BERT(SCR)*, we set its parameters as 21 layers, a word embedding size of 768, and a batch size of 128, all set to their maximum acceptable values in the hardware configuration outlined in section 5.3. In the case of *TAM*, we set its parameters as 12 layers, a word embedding size of 768, a batch size of 96, and a dimension size of 192. These specifications are constrained by the available memory limits.

Notably, when adjusting the parameter size of *BERT(SCR)* to be equal to or greater than *TAM*'s 12 layers, *TAM* consistently outperforms *BERT(SCR)* in terms of accuracy. This result suggests that the superior performance of *TAM* is not solely attributed to parameter size but is also a result of the effectiveness of the Tensorized attention mechanism.

To assess the scalability of *TAM*, we conducted experiments with a reduced number of layers set to 8, as shown in Table 10. The results demonstrate that as the number of layers increases from 8 to 12, the accuracy of *TAM* improves, suggesting a positive impact of scaling up the number of parameters on *TAM*'s performance.

## A.4 COMPUTATIONAL PERFORMANCE DETAILS

In the realm of response selection models, BERT-FP (Han *et al.*, 2021) stands out as state-of-the-art across various datasets. Since BERT-FP is built upon BERT, the integration of TAM with BERT-FP is a straightforward process. While BERT has slightly fewer parameters and marginally faster computation times than TAM in scenarios with shorter context-response lengths, as exemplified in the NFL dataset (Pretraining per epoch: BERT 19 minutes, TAM 33 minutes; Finetuning per

Table 10: Evaluating accuracy and parameter size with varied number of layers in *TAM*.

| Method | NFL | | | | Politics | | | |
|---|---|---|---|---|---|---|---|---|
| Num of layers | $R_{10}$@1 | $R_{10}$@2 | $R_{10}$@5 | #params | $R_{10}$@1 | $R_{10}$@2 | $R_{10}$@5 | #params |
| 8 | 34.59 | 51.54 | 80.22 | 151,488,713 | 41.95 | 58.25 | 84.16 | 151,488,933 |
| 12 | 45.61 | 61.86 | 85.57 | 156,228,041 | 51.51 | 65.92 | 87.65 | 155,047,461 |

epoch: BERT 24 minutes, TAM 27 minutes; Memory: BERT 20,781 MiB, TAM 25,381 MiB), TAM consistently demonstrates superior accuracy within a realistic timeframe. This superiority is evident from the results presented in Tables 2 and 3. Furthermore, TAM is faster than BERT, especially in longer context-response scenarios, as demonstrated by the POLITICS dataset (Pretraining per epoch: BERT 33 minutes, TAM 29 minutes; Finetuning per epoch: BERT 49 minutes, TAM 46 minutes; Memory: BERT 51,531 MiB, TAM 62,817 MiB).

Moreover, alternatives such as RoBERTa (Liu *et al.*, 2019) are available if we explore Transformer Encoder models for response selection except for BERT. However, when applying TAM to RoBERTa, the effects on computational time and memory efficiency are expected to be similar to those on BERT since the fundamental Transformer architecture is shared among them.

