# OpenReview forum: "TENSORIZED ATTENTION MODEL"
_ICLR.cc/2024/Conference — Submitted to ICLR 2024_

### Official Review · Reviewer_4BZc · 2023-10-25

**Soundness:** 3 good
**Presentation:** 3 good
**Contribution:** 2 fair
**Rating:** 5
**Confidence:** 3

**Summary:**

This paper introduces the Tensorized Attention Model (TAM), which leverages Tucker decomposition to calculate attention weights across various object types and seamlessly integrates them into the Transformer encoder. The authors evaluate TAM using the Reddit dataset and demonstrate that it significantly outperforms existing Transformer-based methods in terms of accuracy in response selection tasks.

**Strengths:**

1. The introduction of the Tensorized Attention Model (TAM) is a novel extension to the Transformer model that incorporates multi-dimensional attention mechanisms, enriching attention outputs by considering relationships across three or more distinct object types.

2. TAM innovates the tensorized transformer framework by employing Tucker decomposition for multi-object attention, enhancing accuracy through query-length-aligned tensor decomposition of key and value components, and reducing memory usage and overfitting through iterative averaging while maintaining accuracy.

3. The paper provides empirical validation of TAM's effectiveness by integrating it into a Transformer encoder and evaluating its performance in response selection tasks.

**Weaknesses:**

1. The paper focuses on measuring the impact of TAM's multi-dimensional attention within the encoder model, although TAM could theoretically be applied to both encoder and decoder models.

2. The paper does not provide a detailed comparison of TAM with other state-of-the-art methods in the field, which could help to better understand the advantages and limitations of the proposed approach.

3. The paper does not explicitly mention the application of TAM to other Transformer-based models, such as decoders.

**Questions:**

1. How does TAM compare to other state-of-the-art methods in terms of computational efficiency and memory usage?

2. Are there any potential applications of TAM in other natural language processing tasks, such as machine translation or question-answering?

3. Can TAM be applied to both encoder and decoder models in Transformer-based architectures?

4. What is the potential for scaling up the architecture to larger parameter sizes?

---

> ### Author Response · Authors · 2023-11-22
> **Response#1 to Reviewer 4BZc**
>
> We would like to express our gratitude to the reviewer for your thoughtful comments.
>
> Q1: How does TAM compare to other state-of-the-art methods in terms of computational efficiency and memory usage?
>
> A1: We have included a comparison of computational efficiency and memory usage in the Appendix section titled "A.4 COMPUTATIONAL PERFORMANCE DETAILS" in the revised manuscript.  Kindly Review the Revised Paper in the Attached PDF File. The details are provided below:
>
> In the realm of response selection models, BERT-FP (Han et al., 2021) stands out as state-of-the-art across various datasets. Since BERT-FP is built upon BERT, the integration of TAM with BERT-FP is a straightforward process. While BERT has slightly fewer parameters and marginally faster computation times than TAM in scenarios with shorter context-response lengths, as exemplified in the NFL dataset (Pretraining per epoch: BERT 19 minutes, TAM 33 minutes; Finetuning per epoch: BERT 24 minutes, TAM 27 minutes; Memory: BERT 20781MiB, TAM 25381MiB), TAM consistently demonstrates superior accuracy within a realistic timeframe. This superiority is evident from the results presented in Tables 2 and 3. Furthermore, TAM is faster than BERT, especially in longer context-response scenarios, as demonstrated by the POLITICS dataset (Pretraining per epoch: BERT 33 minutes, TAM 29 minutes; Finetuning per epoch: BERT 49 minutes, TAM 46 minutes; Memory: BERT 51531MiB, TAM 62817MiB).
>
> Moreover, alternatives such as RoBERTa (Liu et al., 2019) are available if we explore Transformer Encoder models for response selection except for BERT.  However, when applying TAM to RoBERTa, the effects on computational time and memory efficiency are expected to be similar to those on BERT since the fundamental Transformer architecture is shared among them.
>
> Q2: Are there any potential applications of TAM in other natural language processing tasks, such as machine translation or question-answering?
>
> A2: QA datasets containing questions, answers, and passages, such as TweetQA \citep{xiong2019tweetqa} and RACE (https://www.cs.cmu.edu/~glai1/data/race/), inherently represent multi-dimensional objects. Moreover, entities like Wikipedia and tweets (currently X) naturally exhibit multi-dimensional characteristics, and TAM can assimilate their complexities, as articulated in the introduction section: "two connected sentences in a Wikipedia article share a common topic, and tweets may trigger a chain of replies connected to specific locations."
>
> We also have conducted an evaluation on the TweetQA dataset, confirming the reproducibility of TAM's superiority over other methods, as outlined in Appendix A.2, titled 'Evaluation on QA dataset.' Our responses to all reviewers, titled 'We have assessed the reproducibility of TAM's superiority over other methods using the TweetQA dataset' (available at https://openreview.net/forum?id=28kAFnQZ5V&noteId=XsXgcwkWr8), include evaluations using TweetQA to affirm reproducibility. Your careful consideration in reviewing the responses is greatly appreciated. Please also refer to the attached PDF file for the revised paper.
>
> Q3: Can TAM be applied to both encoder and decoder models in Transformer-based architectures?
>
> A3: Yes, Tensorized attention can be applied to both encoder and decoder models in Transformer-based architectures.
>
> It is essential to note that applying Tensorized attention to the decoder involves surpassing the accuracy achieved by Pretrained Large
> Language Models (LLMs). This application is envisioned as part of the fine-tuning process on these Pretrained LLMs. We are actively
> exploring additional techniques, such as LORA (refer to [1]) or Prefix tuning (refer to [2]), to enhance the adaptation of Tensorized attention during the fine-tuning stage on Pretrained LLMs. However, due to the potential extensive content and breadth of such research
> explanations, coupled with variations in evaluation metrics, we are considering presenting these findings as a distinct contribution in a
> separate paper. This decision aims to uphold clarity and maintain focus within our current manuscript.
>
> 1. Edward J. Hu, Yelong Shen, Phillip Wallis, Zeyuan Allen-Zhu, Yuanzhi Li, Shean Wang, Weizhu Chen: LoRA: Low-Rank Adaptation of
> Large Language Models. CoRR abs/2106.09685 (2021)
>
> 2. Xiang Lisa Li, Percy Liang: Prefix-Tuning: Optimizing Continuous Prompts for Generation. ACL/IJCNLP (1) 2021: 4582-4597

---

> > ### Author Response · Authors · 2023-11-22
> > **Response#2 to Reviewer 4BZc**
> >
> > Q4: What is the potential for scaling up the architecture to larger parameter sizes?
> >
> > A4: Presently, our experiments are conducted within the hardware configuration outlined in the METRICS section, featuring parameters
> > such as a number of layers of 12, word embedding size of 768, batch size of 96, and dimension size of 192. These specifications are capped by the maximum limits of the available memory. To explore the scalability, we have included experiments in the Appendix A.3 with a reduced number of layers set to 8. The results demonstrate that as the number of layers increases, the accuracy of TAM improves, suggesting that scaling up the number of parameters positively impacts TAM's performance.
> >
> > It's noteworthy that while increasing the word embedding size could potentially enhance scalability, we have kept it constant, as it often depends on the pretrained word embedding model. Additionally, our observations indicate that batch size has a minimal impact on performance. However, concerning the dimension size $D$ of core tensor, there is a trade-off. Assuming a low-rank tensor decomposition, increasing the dimension size up to a certain extent can improve accuracy. Nevertheless, if the dimension size becomes too large, approaching the scale of word embedding size (e.g., 768), it may no longer satisfy the assumption of a low-rank tensor, resulting in a degradation of accuracy.
> >
> > We have included the above observations in the Appendix section titled "A.3 PARAMETER SIZE ANALYSIS: TAM VS. BERT IN LAYER VARIATION ABLATION STUDY" in the revised manuscript.

---

> > > ### Author Response · Authors · 2023-11-23
> > > **Kind reminder**
> > >
> > > Dear Reviewer 4BZc,
> > >
> > > We wish to express our sincere gratitude once again to you for the valuable contributions and considerate feedback. We would like to gently bring to your attention that the discussion phase between authors and reviewers is nearing completion (within 6 hours).
> > >
> > > Given the inclusion of the new experiments and further clarifications, we kindly inquire whether the reviewers might reconsider the evaluation of our submission. Should you have any further insights to share, we are more than willing to sustain our discussion until the deadline.

---

### Official Review · Reviewer_qc5V · 2023-10-31

**Soundness:** 3 good
**Presentation:** 3 good
**Contribution:** 3 good
**Rating:** 6
**Confidence:** 4

**Summary:**

The article presents a new attention mechanism for transformer architectures based on a tensor framework. The proposed tensor based method can incorporate multi-object relationships using Tucker decomposition. In particular, for the attention layer, along with the query Q and memory (key) K embeddings, a semantics embedding S is considered, and the attention mechanism is defined through a Tucker decomposition with a trainable core tensor G. Then an aggregation layer is used to convert the multi-dimensional attention tensor into a 2D matrix by summing up along the semantics axis. Several numerical results are presented an a Reddit dataset to illustrate the performance of the method compared to the standard attention and an alternate tensor attention mechanisms.

**Strengths:**

Strengths:
1. A new tensor based attention mechanism is proposed that can incorporate side information such as semantics and multi-wise interactions into the attention layers.
2. Tensor algebra is a natural approach to define multidimensional interactions and the proposed method modifies existing method to handle multi-way relations.
3. Numerical results show that the method yields promising results and outperforms previous methods.

**Weaknesses:**

Weakness:
1. Intuition behind the use of semantics in the attention layer is not clear.
2. Numerical results are on a single dataset.
3. The method might be incremental.

**Questions:**

The paper presents a tensor approach, that extends the previous work to account for multi-way interactions, and introduces semantics dimension to attention. This might be interesting in applications where there is natural multi-dimensional correlations such as videos, genetics and others.

I have the following comments about the paper:

1.  The intuition behind introducing the semantics information when defining attention, and what information does the 3rd order tensor capture are not very clear. There does not seem to be any activation function (say softmax) used after the Tucker product. Typically, the attention mechanism tries to capture key token to token interactions. Here, it is not clear that does the attention layer learn.


2. The paper presents interesting numerical results. However, there are few questions here.
First, the exposition seems limited as only one dataset is considered with just 2 types of semantics. Are there other datasets or settings/applications where there might be natural multi-dimensional objects.
Second, the evaluation metric considered seems slightly different from other attention based papers,  where typically accuracy is considered. Perhaps R_{10}@1 is similar. Is there a reason why R_{10}@k is considered?
Next, in the results presented, it appears TAM has more #params than the standard BERT. Perhaps the performance gain is due to this. It would be interesting to see if standard attention would come close to TAM if similar #params are used.
Lastly, why does TAM without semantics information perform better than standard BERT or tensorized attention?


Minor Comment:
i. Use \citep to get the standard citation form. Otherwise it results in double parentheses if (\cite{}) is used.

---

> ### Author Response · Authors · 2023-11-22
> **Response#1 to Reviewer qc5V**
>
> We would like to express our gratitude to the reviewer for your thoughtful comments.
>
> Q1: The intuition behind introducing the semantics information when defining attention, and what information does the 3rd order tensor capture are not very clear.
>
> A1: TAM explicitly captures the relationships not only between Query (Q) and Key (K) but also encompasses the relationship with Semantics (S), providing a more comprehensive representation.  Specifically, we construct the Q-K-S tensor by "learning" core tensor \mathcal{G} (The core tensor \mathcal{G} has a $D$-length trainable weight vector \bf{g} primarily on its diagonal), as outlined in equations (4) and (5).  Note that within Equation (5), the softmax function is used in the 's.t.' section.
>
> This extension allows TAM to learn and retain information about how specific tokens in S influence tokens in both Q and K.  The preservation of these relationships is evident despite the flattening of the S axis through semantic fusing, as explained in the qualitative assessment in Figure 2 of the Appendix A.1. In contrast to approaches that concatenate S as a prefix to Q and K (such as BERT(SCR)), our method maintains the independence of the relationships between Q, K, and S, allowing for simultaneous observations. While concatenation approaches may be considered, they inherently lack the ability to independentlyobserve the relationships between Q, K, and S. As a result, they resort to approximate solutions, leading to the observed higher accuracy of TAM compared to BERT(SCR), as demonstrated in experiments and Figure 2.
>
> To clarify the role of S, we have explicitly articulated in the fourth paragraph of the Introduction section of the paper: "These semantic objects represent shared and common semantic elements within query and memory objects." Additionally, we highlighted, "This enables us to consider concurrent occurrences among objects" for further clarity.
>
> Q2: The paper presents interesting numerical results. However, there are few questions here. First, the exposition seems limited as only one dataset is considered with just 2 types of semantics. Are there other datasets or settings/applications where there might be natural multi-dimensional objects.
>
> A2: QA datasets with questions, answers, and passages ( e.g.  TweetQA (Xiong et al., 2019) and RACE (https://www.cs.cmu.edu/~glai1/data/race/)) are natural multi-dimensional objects.  Additionally, Wikipedia and tweets (currently X) naturally have multi-dimensional objects and can be learned by TAM as described in the Introduction section as follows: "two connected sentences in a Wikipedia article share a common topic and tweets may trigger a chain of replies connected to specific locations".
>
> We also have conducted an evaluation on the TweetQA dataset, confirming the reproducibility of TAM's superiority over other methods, as outlined in Appendix A.2, titled 'Evaluation on QA dataset.' Our responses to all reviewers, titled 'We have assessed the reproducibility of TAM's superiority over other methods using the TweetQA dataset' (available at https://openreview.net/forum?id=28kAFnQZ5V&noteId=XsXgcwkWr8), include evaluations using TweetQA to affirm reproducibility. Your careful consideration in reviewing the responses is greatly appreciated. Please also refer to the attached PDF file for the revised paper.
>
> Q3: Second, the evaluation metric considered seems slightly different from other attention based papers, where typically accuracy is
> considered. Perhaps R_{10}@1 is similar. Is there a reason why R_{10}@k is considered?
>
> A3: In the context of response selection, R_{10}@k is commonly considered in many studies, as also cited in our manuscript (e.g., Qian et al.,2021; Han et al., 2021) because it can capture diverse results beyond just the top-ranked response unlike accuracy and R_{10}@1.  It is particularly important to obtain a variety of relevant responses in many applications like IR [1] and Recommendation [2] studies.
>
> [1] Ryan Lowe, Nissan Pow, Iulian Serban, and Joelle Pineau. 2015a. The ubuntu dialogue corpus: A large dataset for research in unstructured multi-turn dia- logue systems. In Proceedings of SIGDIAL.
>
> [2] Makoto Nakatsuji, Yasuhiro Fujiwara, Akimichi Tanaka, Toshio Uchiyama, Ko Fujimura, Toru Ishida: Classical music for rock fans?: novel recommendations for expanding user interests. CIKM 2010: 949-958

---

> ### Author Response · Authors · 2023-11-22
> **Response#2 to Reviewer qc5V**
>
> Q4: Next, in the results presented, it appears TAM has more #params than the standard BERT. Perhaps the performance gain is due to this. It would be interesting to see if standard attention would come close to TAM if similar #params are used.
>
> A4: We have conducted an ablation study by varying the number of layers of BERT, and the results have been summarized in Section A.3 of the Appendix. Interestingly, even when adjusting BERT's parameter size to be equal to or greater than TAM, TAM consistently outperforms BERT in terms of accuracy. This observation suggests that the superior performance of TAM is not solely attributed to the parameter size but is also a result of the effectiveness of the Tensorized attention mechanism.
>
> Q5: Lastly, why does TAM without semantics information perform better than standard BERT or tensorized attention?
>
> A5: It appears there might be a misunderstanding regarding the mention of TAM without Semantic Fusing in Table 4. This variant, as indicated in the text, utilizes the "split&concat" approach, as described in Section 5.5: "However, w/o Semantic Fusing utilizes split&concat to
> compute transformer output directly from the attention weight tensor." This method differs from the approach detailed in Section 4.2, which employs the equation (6) for calculating attention weights between q and k.
>
> The "split&concat" method in TAM without Semantic Fusing reflects a different way of incorporating 3D attention into the transformer output compared to the approach used when Semantic Fusing is applied. Although both "split&concat" and Semantic Fusing leverage semantics, the results indicate that the approach using Equation 6 with Semantic Fusing yields better accuracy than the "split&concat" method.
>
> To prevent the aforementioned misunderstanding, we have revised the text for the ablation study as follows:
>
> For the ablation study, we consider the 'w/o Semantic Fusing' method, employing the same input configuration as TAM: using the query as the dialogue history, the key and value as the current utterance context, and the semantic as the dialogue title. However, 'w/o Semantic Fusing' employs 'split&concat' to compute the transformer output directly from the q-k-s attention weight tensor.
>
>
> Q6: The method might be incremental.
>
> A6: The comment seems to be addressed by comparing our approach with Tensorized Transformer. As summarized in the contributions outlined in Section 1, TAM fundamentally differs by assuming distinct Q, K, and S tensors and substantially improving them as a methodology.  In fact, numerical improvements in Table 2 show the significant advancements of our approach. We view this as a substantial contribution that encourages the evolution of attention models based on tensors.

---

> > ### Author Response · Authors · 2023-11-23
> > **Kind reminder**
> >
> > Dear Reviewer qc5V,
> >
> > We wish to express our sincere gratitude once again to you for the valuable contributions and considerate feedback. We would like to gently bring to your attention that the discussion phase between authors and reviewers is nearing completion (within 6 hours).
> >
> > Given the inclusion of the new experiments and further clarifications, we kindly inquire whether the reviewers might reconsider the evaluation of our submission. Should you have any further insights to share, we are more than willing to sustain our discussion until the deadline.

---

> ### Comment · Reviewer_qc5V · 2023-12-01
>
> I thank the authors for their thorough responses to all reviewers and the changes made to the draft. I wish the authors had not waited till the end of the discussion deadline to provide their responses. Reviewers did not have any time to go over the responses and changes. I am keeping my score.

---

### Official Review · Reviewer_iEpi · 2023-11-07

**Soundness:** 2 fair
**Presentation:** 2 fair
**Contribution:** 2 fair
**Rating:** 5
**Confidence:** 5

**Summary:**

The paper studies the problem of modelling multi-object relationships for attention mechanisms. For this problem, they focus on incorporating object types into attention via proposing tensorized attention which uses Tucker decomposition to acquire attention weights across object types. Experiments on the Reddit dataset verifies the effectiveness.

**Strengths:**

The paper exhibits several strengths:

- The methodology section is clearly written with transparent details.

- In overall, the paper is well-written with few technical errors.

**Weaknesses:**

However, there are some small but significant drawbacks in the paper:

- The logic of the motivation seems confusing. E.g., the authors claim that computing transformer output from attention weights is not suitable for transforming from source object to different target object, but we can calculate multiple attentions for different source-target object pairs via using co-attention [1].

- The argument that BTD leads to overfitting because it uses more than two core tensors seems ad-hoc. It is similar to the argument because previous methods use more parameters, they suffer from overfitting.

- The experiments are incomprehensive. Executing the method on only one dataset is insufficient to assess its effectiveness.

[1] Actbert: Learning global-local video-text representations, CVPR 2020.

**Questions:**

Do you evaluate TAM on other datasets than the Reddit dataset?

---

> ### Author Response · Authors · 2023-11-22
> **Response#1 to Reviewer iEpi**
>
> We first would like to express our gratitude to the reviewer for your thoughtful comments.
>
> Q1:  Authors claim computing transformer output from attention weights is not suitable for transforming from source to different target, but we can calculate multiple attentions via co-attention like Actbert.
>
> A1: We assert that it would reduce accuracy to directly compute transformer output from the attention weight tensor itself, as did in the previous Tensorized Transformer (Ma et al. (2019)). In fact, we say that "Secondly, the approach directly computes
> transformer output from attention weight tensors and thus is inadequate to obtain transformations from source object to a different type target object, making it less versatile" in Section 1.
>
> Contrary to the previous approach, as clearly articulated in the last paragraph of Section 4.2 and expressed in Equation (6), we acquire query-key attention matrix obtained through semantic fusion from the attention weight tensor, and then compute the dot product between query-key attention matrix and the value vector.  As a result, TAM learn the transformation from the source (value from memory) to the target (query) as did in the original transformer.  The co-attention study presented by the reviewer employed a similar approach.
> Specifically, after calculating attention weights between the source and target, the co-attention approach learns the transformation from the source (value from memory) to the target (query), following the original transformer framework.
>
> However, in the case of the previous Tensorized Transformer (Ma et al. (2019)), the model directly outputs transformer output by
> "linearly" mapping the split and concatenated matrix made from query-key-value attention tensor. This approach does not include the step of taking the dot product between the query-key attention matrix and the value vector to transform the distribution. We contend that such a direct mapping results in a degradation of accuracy since it does not learn the transformations from source objects to target objects, and we have conducted experiments to support this claim.
>
> We appreciate the introduction of Actbert.  Actbert enhances encoding from three objects: action, regional object, and linguistic features by blending action features from linguistic representations and guided action features from regional object representations. It calculates source-target attentions between these blended or guided features and each source object (action, linguistic, and regional).  In contrast, TAM emphasizes simultaneous observation of all three objects to overcome the attempt to approximate the relationships of the original three objects into pairs, preventing a failure in accurate relationship analysis. TAM outperforms existing methods like BERT(SCR), as illustrated in Figure 2 in the Appendix A.1, where BERT(SCR) fails to predict events involving the simultaneous observation of three distinct objects. This demonstrates the effectiveness of our proposed approach over approximate methods that compute transformer output by combining two objects at a time.  We have succinctly highlighted the distinctions from Actbert in the related work section.   Kindly Review the Revised Paper in the Attached PDF File.
>
> Q2: The argument that BTD leads to overfitting because it uses more than two core tensors seems ad-hoc.
>
> A2: The reviewer's comment could pertain to a sentence in the introduction of the RELATED WORK section, specifically, "Furthermore, evaluation results show that BTD incurs higher memory overhead. Increasing core tensors beyond two leads to overfitting, while a single core tensor may reduce accuracy."  Particularly, the sentence of the RELATED WORK section might be confusing that "BTD incurs higher memory overhead". Therefore, we have revised the statement as follows: "Increasing core tensors beyond two leads to higher memory overhead and overfitting, while a single core tensor may reduce accuracy."
>
> As highlighted in Table 2, our proposed method, TAM, effectively improves accuracy while maintaining a reduced parameter size compared to the Tensorized Transformer.  Additionally, on the NFL dataset, despite TAM and Tensorized Transformer(CR) having similar parameter sizes, TAM significantly enhances accuracy by using "Semantic fusing" and "query aligned", as evident in Table 4. This suggests that TAM can effectively learn with a smaller parameter size compared to the Tensorized Transformer.
>
> Moreover, as explained in Section 5.6, despite increasing the number of cores, TAM barely increases the parameter size by exploiting the memory-efficient Tucker decomposition.  Additionally, the Appendix A.3 demonstrates that TAM's parameter size increases with Transformer encoder layer size, and in tandem, accuracy improves as shown in Table 10. This implies that TAM does not suffer from overfitting.

---

> > ### Author Response · Authors · 2023-11-23
> > **Response#2 to Reviewer iEpi**
> >
> > Q3: Do you evaluate TAM on other datasets?
> >
> > A3: We have conducted an evaluation on the TweetQA dataset, confirming the reproducibility of TAM's superiority over other methods, as outlined in Appendix A.2, titled 'Evaluation on QA dataset.' Our responses to all reviewers, titled 'We have assessed the reproducibility of TAM's superiority over other methods using the TweetQA dataset' (available at https://openreview.net/forum?id=28kAFnQZ5V&noteId=XsXgcwkWr8), include evaluations using TweetQA to affirm reproducibility. Your careful consideration in reviewing the responses is greatly appreciated. Please also refer to the attached PDF file for the revised paper.

---

> > > ### Author Response · Authors · 2023-11-23
> > > **Kend reminder**
> > >
> > > Dear Reviewer iEpi,
> > >
> > > We wish to express our sincere gratitude once again to you for the valuable contributions and considerate feedback. We would like to gently bring to your attention that the discussion phase between authors and reviewers is nearing completion (within 6 hours).
> > >
> > > Given the inclusion of the new experiments and further clarifications, we kindly inquire whether the reviewers might reconsider the evaluation of our submission. Should you have any further insights to share, we are more than willing to sustain our discussion until the deadline.

---

### Author Response · Authors · 2023-11-22
**To all the reviewers: we have assessed the reproducibility of TAM's superiority over other methods using the TweetQA dataset.**

We first would like to express our gratitude to the reviewers for their thoughtful comments.

We conducted an evaluation on the TweetQA dataset and verified the reproducibility of TAM's superiority over other methods, as detailed
in Appendix A.2, titled "Evaluation on QA dataset." Kindly review the Revised Paper in the Attached PDF File.

We conducted the evaluation on the Tweet QA dataset (Xiong et al., 2019). It is a large-scale dataset designed for automated question
answering, permitting abstractive responses over social media content. It specifically emphasizes meaningful tweets employed by
journalists in the creation of news articles. It consists of question-passage-answer triples. The passages are tweets that have
been used by journalists in news articles, implicitly implying that such tweets contain useful information. It includes 10,692 training
triples and 1,979 test triples. The average question length is 6.95, and the average answer length is 2.45.

We prepared the following three-object relationships as we did in Section 5.1: "The entire topic of a passage'', which is derived from
the word embedding stream for the passage and can serve as ${\bf{Q}}$. "The context of the passage-answer pair'', which is
created by applying GRU (Cho et al., 2014) to the word embedding streams in both the passage and the answer. This is represented as
either ${\bf{K}}$ or ${\bf{V}}$. ``The question assigned for the passage'', which is obtained from the word embedding stream for the
question and represented as ${\bf{S}}$.

We set the passage-answer length to 48 and the question length to 15 for this dataset. These settings cover 95\% of the passage-answer pairs and questions in both datasets. Pre-training was performed for 100 epochs, and fine-tuning was done for 20 epochs on both datasets for all methods since all methods converge until the above epochs.

The results when learning from scratch are present in Table 7. The tendency of the results is almost the same as those of the Reddit
dataset. Interestingly, TAM achieves much higher accuracy than other methods, including BERT(SCR) and Tensorized Transformer(SCR), compared to the improvements observed in the Reddit dataset. We note that BERT(SCR) and Tensorized Transformer(SCR) use the same inputs (e.g., questions, passages, and answers). They concatenate questions with passage-answer pairs in a similar way as done in the Reddit dataset in Section 5.2.  We attribute this significant improvement to the stronger associations among questions and passage-answer pairs, in contrast to the relationships observed among titles and utterances within the Reddit dataset. We also consider that effective utilization of multi-objective relationships in a smaller-sized dataset like TweetQA is essential, as it maximizes the use of the observed limited dataset, in comparison to the Reddit dataset. TAM can effectively learn the co-occurrences of multi-object relationships such as those among tokens in ${\bf{Q}}$, ${\bf{K}}$, and ${\bf{S}}$ from the restricted amount of observations.

Table 8 demonstrates the effectiveness of our semantic fusing and query-aligned approaches in improving accuracy. Particularly, the
results indicate that the query-aligned approach is successful in enhancing accuracy by focusing on the alignment centered around $Q$
for its relationships with ${\bf{K}}$ and ${\bf{S}}$ in learning the core tensor. This is believed to occur because the explicit token embedding of the Query can incorporate the latent semantics derived from the correlation between $q$, $k$, and $v$ in an explicit manner.

Also, as indicated in Table 1, while NFL and Politics datasets originate from Reddit, they exhibit significantly different topics and data distributions. The Politics dataset tends to have longer utterances and titles than NFL, resulting in larger tensor sizes due to increased context, response, and semantic lengths.  By using NFL and Politics datasets, the proposed method, TAM, demonstrating the superiority for the datasets with distinct characteristics.

We acknowledge the reviewer's concerns, and we are confident that our comprehensive evaluation strategy, which includes experiments on different datasets, adequately addresses the issue of incompleteness.

---

### Meta-Review · Area_Chair_o2JB · 2023-12-10

**Metareview:**

In this paper, the authors introduce the Tensorized Attention Model (TAM), which employs Tucker decompositions with multi-core tensors for memory-efficient computation of attention weights across diverse object types, seamlessly integrating them into the Transformer encoder. The experiments on the Reddit and TweetQA datasets show the effectiveness of the proposed method.

In general, the paper is well-written and the experimental results are good. The reviewers raised several concerns about the motivation of the work, incomprehensive experiments, and intuition of the proposed method. The authors tried to address them during the rebuttal, but the reviewers were not fully satisfied with the answers.

**Justification For Why Not Higher Score:**

The reviewers raised several concerns about the motivation of the work, incomprehensive experiments, and intuition of the proposed method. The authors tried to address them during the rebuttal, but the reviewers were not fully satisfied with the answers.

**Justification For Why Not Lower Score:**

N/A

---

### Decision · Program_Chairs · 2024-01-16

Reject